# Machine Learning for Prediction of Immunotherapy Efficacy in Non-Small Cell Lung Cancer from Simple Clinical and Biological Data

**DOI:** 10.3390/cancers13246210

**Published:** 2021-12-09

**Authors:** Sébastien Benzekry, Mathieu Grangeon, Mélanie Karlsen, Maria Alexa, Isabella Bicalho-Frazeto, Solène Chaleat, Pascale Tomasini, Dominique Barbolosi, Fabrice Barlesi, Laurent Greillier

**Affiliations:** 1Computational Pharmacology and Clinical Oncology (COMPO) Unit, Inria Sophia Antipolis–Méditerranée, Cancer Research Center of Marseille, Inserm UMR1068, CNRS UMR7258, Aix Marseille University UM105, 13385 Marseille, France; melanie.karlsen@inria.fr (M.K.); maria.alexa@ensae.fr (M.A.); bicalhoisabella@gmail.com (I.B.-F.); dominique.barbolosi@univ-amu.fr (D.B.); laurent.greillier@ap-hm.fr (L.G.); 2Multidisciplinary Oncology and Therapeutic Innovations Department, Assistance Publique—Hôpitaux de Marseille, Aix Marseille University, 13005 Marseille, France; g-mat@hotmail.fr (M.G.); solene.chaleat@ap-hm.fr (S.C.); pascale.tomasini@ap-hm.fr (P.T.); 3Thoracic Oncology Department, Aix Marseille University, CNRS, INSERM, CRCM, 13385 Marseille, France; Fabrice.BARLESI@gustaveroussy.fr; 4International Center of Thoracic Cancers, Gustave Roussy Cancer Campus, Université Paris-Saclay, 94805 Villejuif, France

**Keywords:** blood counts, lung cancer, response, survival, prediction, machine learning

## Abstract

**Simple Summary:**

We studied determinants of response to immune-checkpoint inhibition in advanced non-small cell lung cancer patients. Specifically, we evaluated the association with response of multiple simple pre-treatment blood markers available from routine examination. We first used classical statistical tools and then developed a machine learning algorithm for individual predictions. We obtained a 69% accuracy. Hemoglobin levels and performance status were the strongest predictors. Neutrophil-to-lymphocyte ratio was also associated with outcome. A benchmark of 8 machine learning models also evidenced that the best model performed almost equally well than a logistic regression (basic statistical learning model).

**Abstract:**

Background: Immune checkpoint inhibitors (ICIs) are now a therapeutic standard in advanced non-small cell lung cancer (NSCLC), but strong predictive markers for ICIs efficacy are still lacking. We evaluated machine learning models built on simple clinical and biological data to individually predict response to ICIs. Methods: Patients with metastatic NSCLC who received ICI in second line or later were included. We collected clinical and hematological data and studied the association of this data with disease control rate (DCR), progression free survival (PFS) and overall survival (OS). Multiple machine learning (ML) algorithms were assessed for their ability to predict response. Results: Overall, 298 patients were enrolled. The overall response rate and DCR were 15.3% and 53%, respectively. Median PFS and OS were 3.3 and 11.4 months, respectively. In multivariable analysis, DCR was significantly associated with performance status (PS) and hemoglobin level (OR 0.58, *p* < 0.0001; OR 1.8, *p* < 0.001). These variables were also associated with PFS and OS and ranked top in random forest-based feature importance. Neutrophil-to-lymphocyte ratio was also associated with DCR, PFS and OS. The best ML algorithm was a random forest. It could predict DCR with satisfactory efficacy based on these three variables. Ten-fold cross-validated performances were: accuracy 0.68 ± 0.04, sensitivity 0.58 ± 0.08; specificity 0.78 ± 0.06; positive predictive value 0.70 ± 0.08; negative predictive value 0.68 ± 0.06; AUC 0.74 ± 0.03. Conclusion: Combination of simple clinical and biological data could accurately predict disease control rate at the individual level.

## 1. Introduction

Immune checkpoint inhibitors (ICIs) are now a therapeutic standard in several advanced cancers, particularly in stage IV non-small cell lung cancer (NSCLC) without genetic alteration [1,2]. The development of ICIs is leading to treat an increasing number of patients with these expensive drugs. Even if the overall response rate is higher with ICIs than chemotherapy, it is equal to about 20% for ICIs in monotherapy [2]. Consequently, there are still 4 patients out of 5 with no response to single agent ICI. Thus, identification of predictive markers for ICIs efficacy is an important unmet medical need.

Biologically, ICIs mechanism of action rely on the immune system and tumor micro-environment. Tumor-infiltrating lymphocytes (TILs) are known to have different effects on survival [3]. Blood counts may be a surrogate marker of these TILs and reflect inflammation and adaptive immune response in lung cancer [4]. In this respect, analysis of blood counts before the start of ICIs showed some interesting correlation with response. In a meta-analysis [5], particularly in melanomas treated with Ipilimumab, a higher lymphocyte count and relative lymphocyte count predicted better overall survival (OS), as for a higher eosinophil count and lower neutrophil count. Neutrophil to lymphocyte ratio (NLR) at baseline has also been investigated, and its decrease was associated with better OS, progression-free survival (PFS) and response [6]. The derived NLR (dNLR = absolute neutrophils count/[white blood count—absolute neutrophils count]) is another ratio which has already been an alternative to NLR in melanomas [7,8] and metastatic colorectal cancer [9]. Furthermore, for lung cancer, in a Chinese meta-analysis published in 2016 [10], high levels of platelet to lymphocytes ratio (PLR) at baseline were associated with poor OS and PFS, but in all types of treatment. Specifically for NSCLC treated by ICIs, a study showed that a score combining dNLR greater than 3 with elevated LDH was correlated with worse outcome for ICIs [11]. Furthermore, an Italian study in NSCLC patients treated by ICIs [12] showed that low NLR and low PLR at baseline were associated with development of immune related adverse events (IRAEs), and low NLR was associated with better outcomes (OS, PFS). However, a comprehensive analysis of all classical blood markers for prediction of efficacy in a large number of patients is still lacking [13].

In the era of precision medicine, machine learning (ML) has recently developed as an alternative to classical statistical analysis [14]. The main difference is that statistical analysis focuses on inference and association between variable(s) and outcome(s), while ML puts emphasis on predictive performances only [15]. In oncology, ML has demonstrated great success for prediction from large-dimensional ‘big’ data, such as genomics or imaging [14]. Nevertheless, such data science methods also have relevance to establish predictive models from smaller sets of variables [16]. In addition, successes have mostly been limited to diagnosis and prognosis but seldom for predictive applications in a clinical oncology context (i.e., for therapeutic decision).

We hypothesized that ML could be useful to accurately predict the efficacy of ICIs in NSCLC patients. The present study aimed to develop a ML model for the selection of patients which could benefit from treatment with ICIs, from simple clinical and biological data.

## 2. Materials and Methods

### 2.1. Patients

In this observational monocentric retrospective study, we analyzed data from all patients older than 18 years of age who were diagnosed with advanced NSCLC and who received at least 1 cycle of ICI (anti-PD-L1, anti PD-1 or anti-CTLA-4), alone or in combination (anti-PD(L)1 and anti-CTLA4) following a first cycle of non-ICI systemic therapy from 2 April 2013 to 14 February 2017. Patients were treated according to available guidelines.

The study protocol and retrospective data collection were approved by the Institutional Review Board of the French Society of Respiratory Diseases (Société de Pneumologie de Langue Française—SPLF), under reference number: CEPRO 2019-007 and patients have signed an informed consent.

### 2.2. Data

Data were retrieved from electronic patient records. Clinical and epidemiological data (age, gender, tobacco status, asbestos exposure, performance status, body mass index), disease characteristics (histology, mutation status, TNM stage), treatment data (type, treatment line, toxicity), biological data (last blood count before first infusion of ICI) and outcome data (response and survival) were collected. From the pre-treatment blood counts, we calculated the PLR, NLR and derived NLR (dNLR) = absolute neutrophils count/(white blood count − absolute neutrophils count). Performance status was dichotomized into <2 or ≥2. 

Tumor response was assessed once every 2 months through computed tomography scans, according to the Response Evaluation Criteria in Solid Tumors, version 1.1 [17]. Definition of response here is the best response observed. Overall response rate (ORR) included patients with complete or partial responses. Disease control (DCR) included patients with complete, partial responses or stable disease. Overall survival was defined as the time from start of immunotherapy to death from any cause, censored at the date of last follow-up. Progression-free survival was defined as the time from start of immunotherapy to documented disease progression or death from any cause, censored at the date of last follow-up period.

### 2.3. Statistical Analysis

In exploratory data analysis, two-tailed Student’s *t*-tests were used for continuous variables and chi-squared tests for categorical variables. Association of clinical and biological data response was assessed using univariable and multivariable logistic regression, as implemented in the *glm* function of the *R* software (version 3.6) [18]. Survival analyses of OS and PFS were performed using univariable and multivariable proportional hazard Cox’s regression models [19,20]. Continuous variables were centered and scaled before these analyses.

### 2.4. Machine Learning 

Feature selection was performed using feature importance given by a random forest classification algorithm applied on the entire data set (*randomForest R* package, no tuning, 1000 trees). Once sorted by mean decrease accuracy, incremental logistic regression models were built with increasing number of variables. The selected optimal set of features was the maximal one before observing a decrease in 10-fold cross-validated accuracy. Predictive machine learning (ML) models were further built and assessed using repeated nested k-fold cross-validation with 5 repeats of 3 outer loops (to assess generalizability) containing each 5 repeats of 3 inner loops (to tune the models). Thus, 15 train and test sets were built to test the predictive performances. The models were implemented under the *tidymodels* framework in *R* version 4.0.4 [21]. They included: logistic regression (*glm*), random forest (*ranger*, 1000 trees), single layer neural network (2600 maximum number of weights), naïve Bayes, k-nearest neighbors (*kknn*, rectangular kernel function) and support vector machines (with linear, polynomial or radial basis kernel function). For each outer fold, models were tuned and trained on the train set and predictions were assessed on the test set. The decision tree was built using the *rpart* engine and tuned for hyper-parameters of tree depth, minimum number of data points required for further splitting (*min_n*) and complexity parameter (*cost_complexity*), using a grid search and a 10-fold cross-validation. The tree was then trained on the entire set.

## 3. Results

### 3.1. Patients and Disease Characteristics 

Overall, 298 patients treated with ICIs for stage IV or relapsed NSCLC were retrieved from our database and analyzed. Patient and disease characteristics are summarized in Table 1. Regarding the treatments with ICIs, 89% (*n* = 266) of patients received an anti-PD-1 antibody, with 96.7% (*n* = 286) patients being pretreated with chemotherapy prior to ICIs. The report of blood counts values at start of treatment with ICIs are reported in Appendix A.

### 3.2. Statistical Analysis

#### 3.2.1. Response

The DCR was 53.4% and ORR was 15.4%. Exploratory data analysis was conducted for association of the considered variables with outcome (Figure 1). Significant associations were found for NLR (*p* < 0.001), derived NLR (*p* < 0.001), hemoglobin (*p* < 0.0001), leukocytes (*p* < 0.01) and neutrophils (*p* < 0.001). These results were confirmed by logistic regression analysis, with additional significance of (Table 2). However, in multivariable analysis, only hemoglobin and PS remained significant. They also remained significant in multivariable analysis controlling for possible additional confounding factors: sex, immunotherapy type and smoking status.

#### 3.2.2. Survival Analysis

Progression-free and overall survival are reported in Appendix A. The median PFS was 3.27 months (95% CI: 2.63–4.07) and median OS was 11.4 months (95% CI: 8.8–15.5). Proportional hazard Cox regression confirmed association of hemoglobin and performance status with response. They were significantly correlated with PFS and OS, in univariable and multivariable analysis (Appendix A). They also remained significant in multivariable analysis controlling for possible additional confounding factors: sex, immunotherapy type and smoking status.

### 3.3. Machine Learning

The ML analysis was conducted for prediction of DCR (classification task) and comprised two steps. First, feature selection and then ML classification. The first step was conducted using random forest-based mean decrease in accuracy (Figure 2A) followed by selection of an optimal number of predictors (Figure 2B). The former revealed hemoglobin level as the strongest predictor of DCR. The second strongest predictor was performance status, followed by NLR. Adding further predictors resulted in a decrease in the cross-validated accuracy of logistic regression models (Figure 2B). Thus, these three variables were selected for further inclusion in ML models.

Multiple machine learning models using this set of variables were then assessed for their predictive abilities of DCR (Figure 3 and Table 3). First, learning curves—which evaluate the predictive abilities of the models with increasing number of patients—demonstrated that convergence to the optimal predictive power had been reached, for each model (Appendix A). Receiver-operator curves were similarly discriminant across the algorithms (Figure 3A), apart from k-nearest neighbors (knn) and the polynomial support vector machine (SVM) with poorer performances. Aside knn, mean areas under the curve ranged 0.72–0.74 (Table 3). Similarly, precision (sensitivity)–recall (positive predictive value) curves were comparable (Figure 3B). Best accuracy was 68%, achieved by the random forest model. Sensitivity was generally low (max 0.58, random forest) while specificity was high (0.73–0.94, Table 3). Best positive (naive Bayes, 0.72) and negative (random forest, 0.68) predictive values suggested good predictive power (Table 3). Altogether, the ML algorithms performances suggested the random forest algorithm as the most adequate, achieving highest score in the largest number of them (accuracy, area under the ROC curve, sensitivity and negative predictive value, Table 3) and exhibiting the smallest inter-score variability (Figure 3C).

A random forest algorithm being hard to interpret, we also trained a decision tree algorithm (Figure 3D). This confirmed performance status (<2 versus ≥2) and hemoglobin level (optimal threshold 13 g/dL) as the most important predictive variable. Then NLR, consistently with our random forest-based importance analysis. This tree could be useful for clinical decision at bedside. For instance, patients with performance status <2 and hemoglobin ≥13 g/dL are predicted to have an 83% chance of disease control.

## 4. Discussion

The selection of patients who will benefit from ICIs therapies is crucial in the era of precision medicine, in order to develop new strategies for those patients who are not likely to respond from current strategies. Clinical examination and blood counts are easily acquired, but their predictive power in combination remains to be determined. While classical statistical methods are appropriate and have been employed for determination of association with outcome [11,22], ML techniques are more adapted for prediction tasks. Therefore, we decided to analyze our data with the help of such methods.

Although well developed in several areas of science and industry, especially for dealing with ‘big data’, the use of ML for clinical oncology has remained limited to date [14,23]. In particular, very few studies have investigated ML for prediction of response to immune-checkpoint blockade, and none has focused on the predictive value of blood counts [16,24,25,26]. In addition, the main limitation of such studies is the small sample size, despite being a critical determinant to ensure the robustness and generalizability of the results [27]. For instance, Wiesweg et al. had 55 patients in the training set and 36 in the test set [25].

In comparison, we analyzed the data from 298 patients, allowing to have higher statistical power. Training and test sets were thus composed of about 200 and 100 patients in training and tests, respectively. This large number of patients ensured that the models had enough information to learn and predict (Appendix A). Our results even show that 200 patients are sufficient to reach the optimal accuracy, for models with six variables. Importantly, our data was collected from clinical practice (i.e., real-world data), which implies larger heterogeneity but is also more reflective of the reality at bedside [22]. The random forest algorithm emerged as the algorithm with best trade-off over all the metrics considered, resulting in a 68% mean accuracy and 0.74 mean area under the ROC curve, on test sets. Nevertheless, logistic regression, single layer neural network and naïve Bayes models performed almost equally well, suggesting that similar predictive power can be achieved with a standard, linear statistical learning algorithm such as logistic regression. This is in line with a review that suggested no benefit of ML over logistic regression for clinical prediction models [28]. 

Associations between blood count and efficacy were consistent with previous studies, in particular correlations with hemoglobin, NLR, dNLR and PLR, were the same than in previous studies [5,6,10,11,29]. This consistency comforts the interest of pre-treatment blood counts for prediction of ICIs efficacy. Neutrophils and leukocytes levels were also associated and predictive of response, a finding from our study that has not been reported elsewhere, to our knowledge. We also demonstrated that an ECOG performance status ≥2 was significantly associated with response, a result that remained to be fully established [30,31]. High PS and a high NLR were correlated with worse outcomes, consistently with the physio-pathological immune hypothesis of peripheral lymphocytes stimulation with ICIs, which could lead to redirect TILs to destruct tumor cells through an activation cascade.

The present study focused on the predictive power of blood biomarkers, but the model could include additional variables, even if their predictive values are not perfect individually. PD-L1 expression [32], because of its discordant results, its heterogeneity among histological components [33] and its poor accuracy when assessed in peripheral blood [34], is not an ideal predictive biomarker but would probably increase the performance of our model. Some characteristics based on genomic alterations could also be added. Tumor mutational burden (TMB), defined as the total number of nonsynonymous mutations in the tumor exome [35] and evaluated by next-generation-sequencing (NGS), seems to be a potential biomarker [36]. Even if its measure has some limitation such as high cost, large input DNA amount needed and time of analysis, lung cancer is among tumors with the highest TMB [37], which correlates with ICI efficacy in NSCLC [38]. Importantly, our results compare favorably with the latter study, where predictive power of (durable) response from PD-L1 expression (AUC = 0.646) or TMB (AUC = 0.601) was inferior. K-RAS mutation is a frequent genetic alteration with contrasting implications for ICIs efficacy. It was initially not linked with poor response to ICIs [39], but a better response when mutated was shown in 2019 [40], whereas the co-occurring genomic alteration of K-RAS and STK11/LKB1 was associated with a primary resistance to PD1 axis blockade [41]. Other recent work using metabolomics data obtained from blood sampling demonstrated impressive predictive power of response, although with small sample sizes [42,43,44]. The integration of such markers in our model might improve its efficacy. Based on the combination of quantitative imaging and machine learning, radiomics study have also started to emerge for prediction of ICIs efficacy from radiological images [24,45]. Such studies should nevertheless be carefully evaluated for their added predictive power in comparison to simpler biomarkers as the ones used here [46]. 

The first limitation of our study is the retrospective design and the limited number of centers involved (2 centers). Another limitation is the lack of tumor PD-L1 assessment. This testing was not systematically performed from 2013 to 2017 as nivolumab in second-line or more did not require the PD-L1 status to be prescribed to NSCLC patients. This status could have helped our algorithm using a quantitative variable, freeing ourselves from a qualitative threshold as currently used: 1% for the refund of Durvalumab in France for the maintenance after a radio-chemotherapy or pembrolizumab in second-line therapy [47] and 50% for the use of Pembrolizumab in first-line therapy [2]. Nevertheless, the PD-L1 determination methods were not the same as nowadays, involving either a different or biased analysis or a new determination of PD-L1 status. 

Certain confounding factors such as bacterial and viral infection, demographic variables such as race, recent chemotherapy, or the use of corticosteroids before treatment could have modified the blood counts analyzed in this study. These data were not collected and not analyzed in our study because we wanted a simple tool to help clinical practice, which could benefit to the majority of patients. Furthermore, blood counts were not necessarily sampled the day of first ICIs infusion, but sometimes one or some days before. This period of time could also have modified the values. On the other hand, our study reflects clinical practice in real-life.

Eventually, the majority of studies to date, including ours, have focused on static pre-treatment predictive markers. However, dynamic markers that integrate on-treatment data start to emerge, with promising predictive value [48]. In such context, the use of statistical Bayesian modeling is particularly appropriate and holds great promise [49]. For ICI, a mathematical model-derived kinetic parameters of tumor kinetics regrowth during relapse has been shown to be the best predictor of overall survival in multivariable analysis including baseline clinical markers [50]. More mechanistic mathematical models for tumor-immune-ICI-radiotherapy dynamics have also been proposed [51]. In light of our results, such models should be applied to include dynamics of blood counts. 

## 5. Conclusions

Blood counts prior to ICIs (elevation of hemoglobin, decrease of NLR, leukocytes or neutrophils) and clinical status (good PS) were significantly associated with better DCR in multivariable analysis. The practical application of these associations using machine learning algorithms was able to predict individual response to treatment. This could be improved further by increasing the number of variables in the model and should be further validated in an independent cohort.

## Figures and Tables

**Figure 1 cancers-13-06210-f001:**
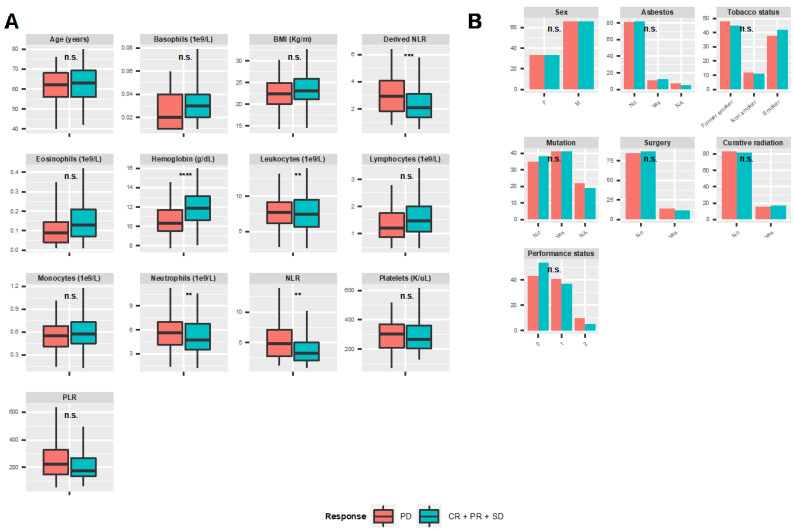
Exploratory data analysis. (**A**) Boxplots of continuous variables. (**B**) Barplots of categorical variables. BMI = body mass index, NLR = neutrophil-to-lymphocyte ratio, PLR = platelets-to-lymphocytes ratio, CR = complete response, PR = partial response, SD = stable disease and PD = progressive disease. Stars indicate statistical significance: **: *p* < 0.01, ***: *p* < 0.001, ****: *p* < 0.0001, n.s. = non-significant.

**Figure 2 cancers-13-06210-f002:**
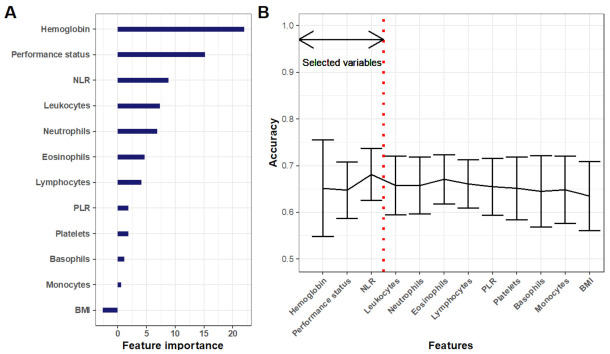
Variable selection. (**A**) Feature importance based on random forest classification and mean decrease in accuracy. (**B**) Accuracy score of incremental logistic regression models built on an increasing number of predictors (i.e., the first one contains only hemoglobin, the second hemoglobin and NLR, etc.). NLR = neutrophil-to-lymphocyte ratio. PLR = platelet-to-lymphocyte ratio. BMI = body mass index.

**Figure 3 cancers-13-06210-f003:**
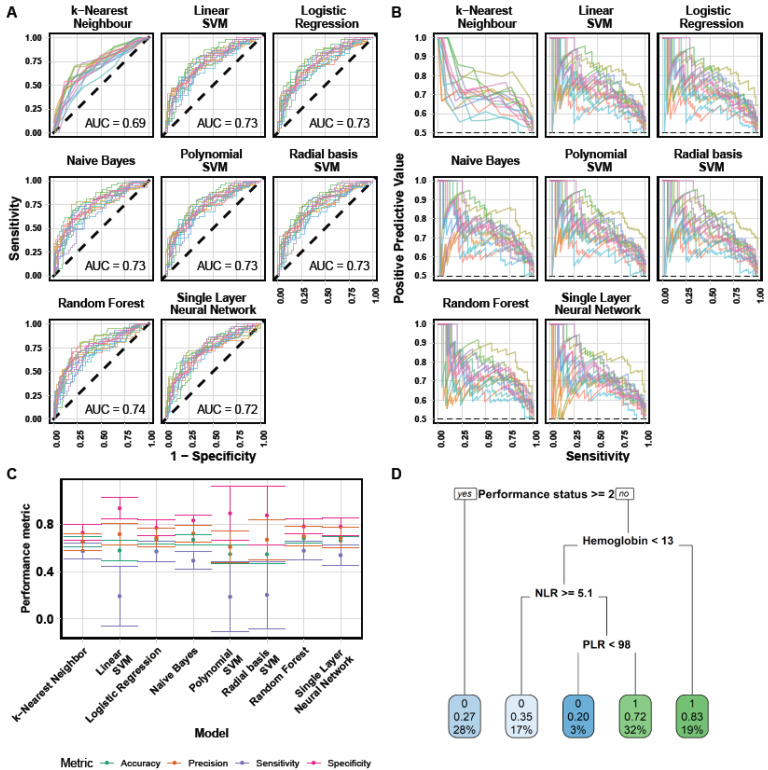
Machine learning algorithms predictive performances. (**A**) Receiver-operator curves for prediction on test sets from each fold of the outer cross-validation loop, for each model. AUC = area under the curve. (**B**) Precision (positive-predictive value)–recall (sensitivity) curves. (**C**) Main performance metrics for each algorithm. (**D**) Decision tree obtained after tuning and training. Each node shows: the predicted class (0 = PD, 1 = CR + PR + SD), the predicted probability of response and the percentage of total observations in the node.

**Table 1 cancers-13-06210-t001:** Patients and disease characteristics.

Variable	N = 298 ^1^
**Age**	62 (55, 69)
Sex	
Female	99 (33%)
Male	199 (67%)
**Tobacco status**	
Former Smoker	140 (47%)
Non smoker	36 (12%)
Smoker	122 (41%)
**Brain metastases**	72 (29%)
**Performance status**	
≥2	26 (8.9%)
0–1	265 (91%)
**Mutation profile**	
ALK	1 (0.8%)
BRAF	9 (7.4%)
EGFR	14 (11%)
KRAS	87 (71%)
Other mutation	5 (4.1%)
ROS1	2 (1.6%)
Wild type	4 (3.3%)
**Immunotherapy type**	
anti-CTLA-4	3 (1.0%)
anti-PD-1	266 (89%)
anti-PD-L1	26 (8.7%)
Combination	3 (1.0%)
**Previous treatment**	
Chemotherapy	281 (95%)
Chemotherapy + immunotherapy	5 (1.7%)
Targeted therapy	11 (3.7%)
**Response**	
Complete response	2 (0.7%)
Partial response	44 (15%)
Progressive disease	131 (45%)
Stable disease	113 (39%)

^1^ Median (inter-quartile range); n (%).

**Table 2 cancers-13-06210-t002:** Logistic regression analysis for disease control.

	Univariable Logistic Regression			Multivariable Logistic Regression		
Variable	Odds Ratio [95% CI]	*p*	Signif	Odds Ratio [95% CI]	*p*	Signif
Lymphocytes	1.1 [0.83, 1.4]	0.678		0.98 [0.15, 5.2]	0.984	
NLR	0.49 [0.31, 0.73]	0.000879	***	0.68 [0.098, 1.9]	0.651	
Platelets	1 [0.82, 1.3]	0.762		1.3 [0.72, 2.4]	0.404	
PLR	0.84 [0.64, 1.1]	0.156		1.1 [0.5, 2.4]	0.788	
Leukocytes	0.68 [0.5, 0.89]	0.00791	**	0.6 [0.0022, 3 × 10^2^]	0.847	
Hemoglobin	1.9 [1.5, 2.5]	9.26 × 10^−7^	***	1.8 [1.3, 2.4]	0.000122	***
dNLR	0.63 [0.47, 0.83]	0.00155	**	0.8 [0.33, 2.7]	0.689	
Neutrophils	0.62 [0.45, 0.83]	0.00232	**	1.5 [0.0047, 2.7 × 10^2^]	0.863	
Monocytes	0.87 [0.69, 1.1]	0.226		0.86 [0.5, 1.4]	0.545	
Eosinophils	1.3 [0.97, 1.9]	0.139		1.1 [0.75, 1.8]	0.582	
Basophils	1.2 [0.95, 1.8]	0.177		1.2 [0.89, 1.8]	0.321	
BMI	1.2 [0.95, 1.5]	0.123		1 [0.76, 1.3]	0.997	
Performance status	0.5 [0.39, 0.64]	6.21 × 10^−8^	***	0.58 [0.44, 0.75]	7.79 × 10^−5^	***

Stars indicate statistical significance: ** : *p* < 0.01, *** : *p* < 0.001. CI = confidence interval. signif = significant.

**Table 3 cancers-13-06210-t003:** Summary of machine learning algorithms predictive performances (mean ± standard deviation, bold entries are maximum values).

Model	Accuracy	ROC AUC	PPV	NPV	Sensitivity	Specificity
Random Forest	**0.68 ± 0.04**	**0.74 ± 0.03**	0.70 ± 0.08	**0.68 ± 0.06**	**0.58 ± 0.08**	0.78 ± 0.06
Logistic Regression	0.67 ± 0.04	0.73 ± 0.03	0.69 ± 0.08	0.67 ± 0.06	0.57 ± 0.09	0.77 ± 0.07
Naive Bayes	0.67 ± 0.04	0.73 ± 0.03	**0.72 ± 0.07**	0.65 ± 0.06	0.49 ± 0.07	0.83 ± 0.05
Single Layer Neural Network	0.66 ± 0.03	0.72 ± 0.03	0.69 ± 0.09	0.66 ± 0.06	0.54 ± 0.09	0.78 ± 0.07
k-Nearest Neighbour	0.66 ± 0.04	0.69 ± 0.04	0.65 ± 0.07	0.66 ± 0.06	0.58 ± 0.07	0.73 ± 0.07
Linear SVM	0.58 ± 0.09	0.73 ± 0.03	0.72 ± 0.09	0.58 ± 0.10	0.19 ± 0.25	**0.94 ± 0.09**
Polynomial SVM	0.55 ± 0.08	0.73 ± 0.03	0.61 ± 0.13	0.58 ± 0.13	0.19 ± 0.29	0.89 ± 0.23
Radial basis SVM	0.55 ± 0.08	0.73 ± 0.03	0.67 ± 0.17	0.56 ± 0.06	0.20 ± 0.28	0.88 ± 0.25

## Data Availability

The data presented in this study are available on request from the corresponding author. The data are not publicly available due to ethical and privacy restrictions.

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
