# Peer review of "Machine Learning for Prediction of Immunotherapy Efficacy in Non-Small Cell Lung Cancer from Simple Clinical and Biological Data"

_cancers, 2021, doi:10.3390/cancers13246210_

Round 1

Reviewer 1 Report

In this study, the authors applied machine learning to haematological data with the aim of predicting the efficacy of immunotherapy in NSCLC patients. A large cohort of patients has been used and different machine learning models were tested.

Among the different methods tested, the Random Forest algorithm provided the best accuracy for the prediction (68%). In general, 68% of prediction accuracy is not so striking. Nevertheless, as the authors stated in the introduction, the overall response rate of immunotherapy is around 20% and today no highly accurate predictive biomarkers of the efficacy exist. Thus, the identification of predictive markers - even if, not with high accuracy but using easily accessible parameters like blood count - is extremely important to help and assist the clinical decision. Indeed, the same approach based on more complex “omics” data, such as metabolomics, provided significantly higher prediction accuracies (see for example doi.org/10.3390/cancers12123574; doi.org/10.1172/jci.insight.133501; doi.org/10.1038/s41467-019-12361-9). This aspect should be better addressed and commented in the discussion of the results, and the above papers quoted.

A better description of the experimental design should be performed. In particular, to be predictive, the blood parameters should be collected before the beginning of the therapy. This aspect is not clearly stated in the materials and methods section, but it only emerges at the end of the discussion.

 The authors could also test the presence of confounding factors, such as the different immunotherapy types, sex, tobacco status. The selection of a smaller but more homogenous group of samples may increase the predictive power of the model. A permutation test may be used to verify the significance of the obtained results.

Reviewer 2 Report

The authors presented an article about the benefit in using machine learning (ML) to predict the response to immunotherapy in non-small cell lung cancer (NSCLC). The paper is well written and logic in its conceptualization. The authors compared the results from both statistical analyses and ML, showing that the latter can be the best approach for prediction of immunotherapy efficacy. The strength of this paper is a good number of patients in the cohort of the study and also the correlation between blood counts and immunotherapy efficacy.

I have some small suggestions.

1) In materials and methods, it is mandatory to state that patients have signed an informed consent to be included in research studies.

2) In materials and methods, I suggest to avoid repeating all the abbreviations that have been already detailed in the introduction (for examples, lines 101 to 110). 

3) In my opinion, should be better to add a "conclusions" part, maybe by moving the last sentences of the discussion part and adding some other information to complete a short paragraph, if possible. 

4) Figure 3D is not much clear, some words are very small. 

Round 2

Reviewer 1 Report

The manuscript has been appropriately revised following my comments and suggestions.